# Impact of e-money on money supply: Estimation and policy implication for Bangladesh

**Ahmed Mehedi Nizam** [ID] *

The Central Bank of Bangladesh, Motijheel, Dhaka, Bangladesh

* ahmed.mehedi.nizam@gmail.com

## Abstract

With the rapid proliferation of mobile telephony and the establishment of an IT-enabled payment and settlement system, Bangladesh nowadays is experiencing a remarkable growth in the usage of mobile financial services (MFS). As more and more people are opting to use this service, a huge number of mobile accounts are opened every day and a substantial amount of money is deposited, withdrawn and transferred frequently through the mobile network. This ever-increasing amount of mobile money flowing through the network may have a sizeable impact on the overall money supply of the country. Thus far, no systematic study has been conducted to quantify the impact of the mobile money on the conventional money supply of Bangladesh. In this study, we attempt to quantify the contribution of mobile money on the money supply which is an important quantity-based nominal anchor of monetary policy in Bangladesh. Apart from deriving algebraic relationships between money supply and e-money, here we have empirically shown that during the 03 years span of 2018-2021, MFS transactions account for nearly 10.88% and 11.29% of total narrow and broad money supply of Bangladesh as on January 2021. Besides, we also qualitatively discuss the impact of e-money on an important price-based nominal anchor of monetary policy in Bangladesh, i.e., interest rate. Based upon the above discussion, here we argue that MFS can act as an effective tool to slash interest rate by a reasonable proportion through adding significantly to the overall supply of money in Bangladesh.

## 1 Introduction

For the last couple of decades, Bangladesh has been experiencing an unprecedented growth of mobile telephony, thanks to the adoption of new technology by the mobile operators in the region. According to a July 2020 report, two out of four mobile operators in Bangladesh have exceeded the 80% landmark of 4G network coverage while the rest of the two are rapidly following the trail [1]. With the expeditious advancement of wireless technology, the number of active mobile users are also rising tremendously. Eventually, as on January 2021, the total number of mobile phone subscribers in Bangladesh has reached nearly 171.85 million [2], which, according to latest available data [3], is nearly 1.04 times the total population of Bangladesh. Moreover, it is interesting to note that this huge number of mobile subscribers are not

**Data Availability Statement:** Data used to generate results are available from the public repository: https://www.bb.org.bd/pub/index.php.

**Funding:** The author(s) received no specific funding for this work.

**Competing interests:** The authors have declared that no competing interests exist.

concentrated in urban areas of the country. Rather, a large number of rural population happen to have mobile connectivity. To be precise, nearly 94.10% of the urban households and 85.20% of the rural households in Bangladesh are seamlessly connected to the mobile network according to the recent government survey [4]. The survey also reveals that 53.20% of the male respondents tend to use mobile internet while 34.20% of women participants happen to do so [5].

Apart from access to mobile telephony, people in some cases of MFS transactions need to subscribe to internet packages in order to unravel the true benefits of MFS. To our surprise, data from Bangladesh Telecommunication Regulatory Commission (BTRC), the telecommunication controlling and bandwidth allocating authority of Bangladesh, unveil that as on January 2021, the total number of mobile internet subscribers reaches an astounding 103.19 million while the number of ISP and PSTN subscribers is next to 9.52 million [6]. Combining the two, Bangladesh apparently has nearly 112.71 million internet subscribers out of its 164.69 million population. So, approximately 68.44% of the Bangladeshi households have access to internet (at least to some extent) which is quite a formidable feat for a least developed country like Bangladesh [7]. These extensive mobile networks and internet coverages throughout the country can be considered as an important stepping stone towards mobile financial inclusion in Bangladesh.

Due to proliferation of mobile telephony and availability of data network around the country, MFS is gaining popularity amongst the retail customers and micro-merchants in Bangladesh. According to a consumer behavior survey conducted in association with UN Capital Development Fund (UNCDF), it is observed that around 30% of the micro-merchants in Bangladesh tend to use mobile financial services to conduct their business [8] which indeed is quite an achievement. In fact, providing financial services to the impoverished segments as well as to the micro-merchants at little to no cost has nowadays become a major enabler of economic development in the developing and the least developed countries [9, 10]. Some studies have even suggested that the absence of an inclusive financial system may add to persistent income inequality and dampened economic growth [9] and Bangladesh is doing quite well in this regard using its ever-growing mobile networks.

With such an extensive market penetration by the mobile operators, mobile financial services can come out to be an effective tool for promoting financial inclusion in rural and slum areas of Bangladesh. According to the latest data released by the Central Bank, as on June 2021, there are nearly 101.24 million mobile accounts operating in Bangladesh of which 58.67% or 59.40 million are rural accounts [11]. During the height of the COVID-19 pandemic, many micro-merchants either partially or fully transitioned to mobile financial services instead of regular cash based transactions and interestingly a significant sum of these micro-merchants turns out to be female [12]. So, apart from making considerable impacts in promoting financial inclusion in the rural Bangladesh, MFS is also working effectively in reducing gender inequality in financial sectors of Bangladesh by empowering woman to some extent [12]. To date, numerous analyses have been conducted to judge the efficacy of MFS in promoting financial inclusion, reducing gender inequality, woman empowerment through MFS merchant accounts and its contribution to overall economic development in the context of Bangladesh. But, to the best of our knowledge, no systematic study has ever been conducted to investigate the role of digital money issued by MFS providers on the overall money supply of the economy in the context of Bangladesh. Here, we argue that the digital money issued by the MFS providers can significantly interrupt total currency in circulation, narrow and broad money supply of the country and we all know that the monetary aggregates, namely, narrow money and broad money are two important quantity based nominal anchors of monetary policy in Bangladesh [13]. In the process of manipulating money supply, e-money issued by the

MFSs can also interact with the price-based nominal anchor of the monetary policies in Bangladesh, i.e., short term nominal interest rate. As the nominal anchors are disrupted, so are the target variables including inflation, unemployment and the alike. In this study, we quantify the changes in currency outside banks as well as the changes in narrow and broad money supply brought about by the digital money issued by the MFS providers and qualitatively discuss its implication on interest rate in light of monetary theory. The rest of the article is organized as follows: Section: 2 provides some preliminary definitions of the terms used in the analysis for clarification of the exposition. Section: 3 briefly discusses the current state of mobile financial services in Bangladesh. Section: 4 illustrates different services provided by the MFSs in Bangladesh and explains their implications on money stocks. Section: 5 provides the methodology used for empirical analysis. Section: 6 elaborates the findings of empirical results. Section: 7 provides a brief discussion and policy implication of the exposition presented here. Section: 8 discusses how the research presented in this piece can be further extended for the better understanding of the dynamics of the MFS in Bangladesh. Finally, Section: 9 makes some concluding remarks.

## 2 Some preliminary definitions

Before delving into detail, a few preliminary definitions are required for clarity of the exposition presented here.

- **Electronic money**: Electronic money, better known as e-money, is a digitized monetary value issued by any licensed MFS provider which represents a claim of the bearer (of the e-money) on its issuer [14]. The issuer of the e-money, i.e., the MFS first receives an equal amount of legal tenders from the intended customers before issuing the e-money. The e-money thus received by the customers is redeemable at the agent points of the respective MFS and can also be used to purchase on different e-commerce/f-commerce sites, utility bill payments, mobile recharge and many more as per the provisions provided by the respective MFS.

- **Trust cum settlement account (TCSA)**: According to Bangladesh Mobile Financial Services (MFS) Regulations 2018 [14], every mobile financial service provider needs to open and operate a trust cum settlement account (TCSA) with a scheduled bank and the balance at that account at any point of time must not be less than the cumulative amount of e-money issued by the MFS [15]. Bangladesh Bank, the central monetary authority of Bangladesh, will oversee the trust account and if any deficiency in the trust account is detected, i.e., the balance in the TCSA of an MFS becomes less than the e-money it has issued, Bangladesh Bank will instruct the trustee of the respective MFS to fill up the gap as early as possible. Moreover, the TCSA must be kept separate from other operational bank accounts of the MFS.

- **Monetary base (MB)**: Monetary base (MB) or base money refers to the total amount of bank notes and coins issued by the central bank that are still in circulation. The bank notes in Bangladesh are issued by Bangladesh Bank against some pre-specified types of assets as indicated in the section: 30(1), clauses (a)-(b) of Bangladesh Bank Order 1972 [16]. Types of assets against which currency can be issued include gold coins, gold bullion, silver coins, silver bullion, Special Drawing Rights (SDR), Asian Monetary Unit, Islamic Dinars, Taka coins issued by the ministry of finance, some permissible types of bill of exchanges and promissory notes as specified in clauses (a)-(b) of section: 30(1) of the said act and most notably, against government guarantees. To see a realistic view of the assets against which paper currency is issued in Bangladesh, one can consult the issue department balance sheet of Bangladesh Bank which is publicly available on a weekly basis [17]. Total amount of currency thus issued by the central bank is known as the Monetary Base (MB). Monetary base (MB) is also frequently

referred to as base money, high powered money, reserve money, outside money, central bank money etcetera and it is this money that eventually gives birth to all the demand and time deposits in the economy through the process of fractional reserve banking. Precisely, monetary base constitutes total currency circulating in public, physical vault cash of the commercial banks and the commercial banks' reserve held in the custody of the central bank [18].

- **Currency outside bank**: Currency outside bank includes all the bank notes and coins that are physically held by the households, companies and all other economic entities at a certain point of time [19]. However, people's demand and time deposits with the depository money banks are not included into currency outside bank as they are illiquid to some extent. Currency outside bank is usually considered as the most liquid form of currencies and can be used upright to make any purchase without the use of any electronic medium like plastic cards, mobile wallets etcetera.

- **Narrow money (M1)**: According to the OECD definition, narrow money (M1) comprises currencies, i.e., banknotes and coins, plus overnight bank deposits [20]. To be precise, it includes currencies outside banks, deposits maintained by the financial institutions with the central bank (both the required reserve and excess reserve) and the total demand deposits of the public maintained with the financial institutions and banks that are revocable on demand without any penalty which is in contrast to time deposits which have specific maturities and cannot be withdrawn before the maturity period without incurring penalties. Thus, currency outside banks is a subset of the narrow money (M1).

- **Broad money**: According IMF's International Financial Statistics, broad money (M2) consists of currency outside bank and total demand and time deposits of the banks and non-bank financial institutions [21]. Thus, the narrow money (M1) is a subset of the broad money (M2). M2 is one of the key economic indicators often used to forecast inflation using the quantity theory of money [22]. And in Bangladesh, it is also one of the quantity-based nominal anchors of the half yearly monetary policy statements formulated by Bangladesh Bank [13].

- **Money multiplier (MM)**: Money multiplier (MM) refers to the ratios of different monetary aggregates like M1, M2 to the total monetary base (MB). In fact, money multiplier represents the number of times the base money has been multiplied through successive savings and lending and this process is formally known as fractional reserve banking. So, it can be estimated by dividing different monetary aggregates like M1, M2, M3 etcetera by the monetary base. Thus, depending upon the monetary aggregates used as a numerator in the estimation of money multiplier, the value and interpretation of the money multipliers (MM) differ. In this study, we define and calculate two different money multipliers, namely, narrow money multiplier ($MM_n$) and broad money multiplier ($MM_b$) and they are defined in the following manner.

$$MM_n = \frac{M1}{MB}$$

$$MM_b = \frac{M2}{MB}$$

## 3 Current state of mobile financial services in Bangladesh

Due to mobility, availability and personalized services without waiting time, mobile financial services are gaining momentum and are spreading rapidly across South Asia, Sub-Saharan

**Table 1. Summarized statistics of MFS activities in Bangladesh for two consecutive months.** Data source: Bangladesh Bank [24].

| Serial no. | Description | Amount in April 2021 | Amount in May 2021 | % Change (April 2021 to May 2021) |
|---|---|---|---|---|
| 1 | No. of Banks currently providing the Services | 15 | 15 | – |
| 2 | No. of agents | 1,061,128 | 1,086,018 | 2.3% |
| 3 | No. of registered clients in Lac | 964.76 | 981.32 | 1.7% |
| 4 | No. of active accounts in BDT Lac | 367.49 | 396.50 | 7.9% |
| 5 | No. of total transaction | 304,978,609 | 346,701,611 | 13.7% |
| 6 | Total transaction in taka(in crore BDT) | 63,478.85 | 71,246.88 | 12.2% |
| 7 | No. of daily average transaction | 10,165,954 | 11,183,923 | 10% |
| 8 | Average daily transaction (in crore BDT) | 2,115.96 | 2,298.29 | 8.6% |

Africa and other developing and least developed countries across the globe [23] and Bangladesh is no exception. However, Bangladesh currently allows only a bank-led MFS model whereby a bank operates MFS services as a product or it may form a subsidiary to do so holding at least 51% of the voting shares of the subsidiary thus formed [14]. In Bangladesh, Mobile Financial Service (MFS) was effectively launched for the first time in March 2011 [25] and to date there are 15 MFS providers working throughout the country rendering a wide range of services including cash-in and cash-out at an extensive number of agent points, facilitating person to person (P2P), business to person (B2P), person to business (P2B) transactions, merchant payments, government payments and inward remittances to name a few [26]. In contrast to the bank and other non-bank financial institutions, today MFS agents are effectively reaching the remotest part of the country, thanks to its portability and low establishment cost. According to the latest available data, the 15 MFS providers are now operating with more than 1 million agent points (both urban and rural) serving an active customer base of over 39 million while the total number of registered clients is as high as 98 million as on May 2021 (see Table 1). Table 1 portrays a comparative position of consolidated MFS operations in Bangladesh for two consecutive months.

From Table 1, it can be seen that every MFS parameter except for the number of MFS providers increases significantly from April 2021 to May 2021 and this is not the case for two months only. Rather, MFS transaction volume, transaction number, number of account holders, number of agent points and all other MFS parameters are increasing tremendously in every single month ever since its inception [24].

From Table 1, it is evident that more than 30 crore MFS transactions take place every month and the total amount associated with it comes close to BDT 60,000-70,000 crore. Meanwhile, the total number of average daily transactions through MFS is found to be greater than one crore with a monetary value higher than BDT 2000 crore Table 1. Thus, we can say that the MFS providers in Bangladesh have successfully penetrated the local market faster and more conveniently than their peers, i.e., banks. In contrast, banks are lagging behind on the national drive of financial inclusion due to their extensive establishment costs and scanty returns for their rural branches. So, from business perspective, the banks are indeed reluctant to open and operate rural branches. As on May 2021, the total number of bank branches in Bangladesh is found to be 10,765 of which 5,594 branches are urban while the rest 5171 are rural [27]. Bangladesh Bank, the central bank of Bangladesh is continuously urging the scheduled banks to open rural branches with a view to bring the unbanked community under the banking umbrella. To do so, Bangladesh Bank has released 'Prudential Regulations for Banks: Selected Issues' in which the scheduled banks are instructed to open new branches in 1:1 ratio, i.e., for every new urban branch, the bank must need to open a rural branch also in order to

get banking license for the newly opened urban branch [28]. But, opening a bank branch requires substantial investment in infrastructure, ICT equipment, network connectivity and necessary staffing. The huge investment thus needed to open a branch in remote places does not usually pay back as the deposits in the rural area are scanty and businesses are virtually non-existent where most of the inhabitants rely on agriculture, cottage industry and personal farming. To effectively circumvent the difficulties of opening and operating rural branches and at the same time to stay in the business, banks have come out with an innovative solution: Instead of opening a full-fledged branch in the rural area, banks rather opt to open some agent points. An agent point is none other than a small banking booth providing limited scale financial services to the unbanked community. To streamline agent banking operations across the country, Bangladesh Bank has issued agent banking guideline for banks back in 2017 [29]. Since then Bangladesh has been experiencing a rapid escalation of agent banking activities specially in the sub-urban and rural areas and as on May 2021, total number of bank agent points and outlets reaches 12,643 and 16,807 respectively [27]. Out of 12,643 agent banking points, 1757 of them are urban while the rest 10,886 are rural. On the other hand, out of 16,807 agent banking outlets, 2167 are urban and 14,640 are rural. So, as on May 2021, the total number of rural bank branches and rural agent banking points and outlets sums up to 30,697. On the contrary, the total number of urban bank branches and urban agent banking points and outlets is found to be 9518 only. However, the total number of MFS agents as on May 2021 is determined to be 10,86,018 Table 1. These figures are quite indicative of the fact that MFS agent points are substantially higher in number than the banks and their agent banking points and outlets combined adding significantly to the continuous process of financial inclusion in Bangladesh.

## 4 Services provided by MFS and their impact on money supply

The following list provides some of the permissible financial services currently offered by the MFS agents in Bangladesh.

- **Cash-in**: One of the most important services rendered by the MFSs in Bangladesh is the cash-in facility. The term cash-in implies the exchange of cash (legal tender) for e-money. Cash-in transaction has a direct impact on currency outside bank, narrow money and in turn the broad money also. When the customer hands over the cash to an MFS agent, the agent receives the cash and issues an equivalent amount of e-money in favor of the customer [14]. At the time of issuance of the e-money, TCSA of the MFS must be raised to reflect the changes in the total amount of e-money issued. Consequently, the MFS increases the balance of its TCSA by the extent of the cash-in transaction. Through cash-in transaction, currency outside bank enters into the banking system, i.e., in the TCSA of the respective MFS which is usually maintained with a scheduled bank as a regulatory requirement. These newly injected monies then eventually get multiplied according to the theory of money multiplier.

- **Cash-out**: Cash-out transaction takes place when an owner of e-money wishes to convert his/her digital balance to legal tenders. To do so, he/she needs to go to some agent points where cash can be withdrawn in exchange of his/her electronic money. Through cash-out transaction, customer liability of the MFS reduces by the extent of the transaction amount and it will eventually gets reflected into the balance of its TCSA [14]. As the customer liability reduces, so does the balance of the TCSA. In cash-out transaction, physical currency comes out of the banking channel and enters into people's pocket and is eventually used to make transactions in cash form. So, due to cash-out transaction, currency outside bank increases while narrow and broad money decreases.

- **P2P transaction**: In P2P transaction, electronic money is transferred from one of the MFS customer accounts to another account and no new e-money is issued or withdrawn during the process. So, the total customer liability for the respective MFS does not change and its balance at TCSA remains the same as before. As no money enters or leaves the banking system, no alteration occurs in any of the three monetary aggregates, namely, currency outside bank, narrow money (M1) and broad money (M2). The beneficiary of the P2P transaction may opt to cash-out his/her money which may alter currency outside bank as well as narrow money (M1) and broad money (M2) also. However, the impact of cash-out transaction by the receiving entity of a P2P transaction has already been accounted for in the consolidated monthly cash-out transaction data. So, at this stage, we no longer need to worry about the impact of probable cash-out by the beneficiaries of the P2P transaction on currency outside bank, narrow money and broad money as it is already taken cared of as part of total monthly cash-out transaction.

- **Salary Disbursement (B2P)**: Companies often prefer to disburse salaries to their employees through B2P transactions using MFS. During this transaction, money is transferred from the company's bank account to MFS's bank account. MFS then stores the money received from the business into its TCSA and issues an equal amount of e-money to be disbursed to the respective recipients. Again, in this transaction, balances in company's bank account are simply transferred to the MFS's TCSA account. Thus, no money enters/leaves the banking channel and all the above three monetary aggregates remain unchanged. However, the employees may choose to cash-out and in the process, they may alter the three monetary aggregates discussed above. Cumulative impacts of all such cash-outs have already been addressed as part of monthly total cash-out transaction.

- **Utility bill payment (P2B)**: P2B transactions are often used by the MFS users for utility bill payments. For the payment of utility bills, e-money in MFS customer's account is debited and utility service provider's bank account is credited. As the MFS customer's account is debited, MFS's external liability reduces and it gets reflected into the balance of its TCSA, i.e., its balance is reduced by the amount of the utility bill. At the same time, utility service provider's bank account is credited. So, total bank deposits remain unaltered and no cash inflow/outflow occurs to/from the banking system. Utility provider may want to withdraw cash from its respective bank account. But, that does not constitute an MFS transaction and should not be a subject of the current study.

- **Merchant payments**: In this transaction, e-money is only transferred from the purchaser's account to merchant's account. So, the volume of total e-money issued does not change. As a result, the MFS's TCSA maintains the same balance as before and no additional bank deposit is created leaving the monetary aggregates totally uninterrupted. If the respective merchant wishes to cash-out, only then currency outside bank, M1 and M2 are changed. All such realized cash-out transactions have already included into the figure of total monthly cash-out transaction.

- **Government payments**: Examples of G2P payments include disbursements of elderly allowances, freedom fighter allowances, agricultural subsidies among others. In this type of transaction, money from government's bank account is transferred to the respective MFS's TCSA account and the MFS then issues an equivalent amount of e-money in favor of the respective beneficiaries. So, the total bank deposits do not change as it is simply transferred from government account to MFS's TCSA and no money leaves or enters the banking channel until and unless the beneficiaries opt to cash-out. Meanwhile, the beneficiaries may opt to cash-out their allowances and in the process, they may alter currency outside bank, M1 and M2.

Like before, the impact of all such cash-out transactions is already accounted for as part of the total monthly cash-out transaction.

- **Inward remittances**: Currently, anyone residing outside of Bangladesh can legally send money to their kith and kin through MFS. However, MFS can only handle such foreign remittances if it is received as credit through the nostro accounts of some scheduled commercial banks operating in Bangladesh under authorized dealer (AD) license [14]. Once the nostro account of the receiving bank is credited, it (the bank) will then hand over an equivalent amount of BDT to the respective MFS. So, the MFS's TCSA is increased while the remittance receiving bank's local currency holding is decreased. Thus, the total volume of local currency deposit in the banking system remains unchanged and so does the local monetary aggregates. However, the remittance receiving bank's foreign currency balance is increased by the size of the current transaction as its nostro account is credited. These nostro accounts are maintained with some foreign banks operating abroad and are part of the money supply of the respective foreign land. Thus, the total money supply of the remittance receiving country remains virtually unchanged after receiving such foreign remittances. Meanwhile, the MFS will then issue an equivalent amount of e-money in favor of the respective beneficiaries. If the beneficiaries choose to cash-out their e-money, only then currency outside banks as well as the other two monetary aggregates are changed accordingly. However, the amount cashed-out by the respective beneficiaries of the foreign remittances during a particular month has already been included into the consolidated monthly cash-out transaction.

The following tables summarize the impacts of the aforementioned transactions on the three monetary aggregates, namely, currency outside bank, narrow money (M1) and broad money (M2). The first table assumes that in a particular month, the total volume of cash-in, cash-out, P2P, B2P, P2B, merchant payment, government payment and inward remittances cleared by the MFS agents throughout the country amount to $A$, $B$, $C$, $D$, $E$, $F$, $G$ and $H$ respectively.

The Table 2 summarizes the impact of different MFS transactions on various monetary aggregates as well as on the consolidated balance of TCSA of all the MFSs operating in the country during a particular month. If we intend to quantify the cumulative impact of mobile financial transactions on different monetary aggregates over the months 1 to $n$, then we get Table 3 instead. In Table 3, we have subscripted every quantity, i.e., $X_i$ implies the value of a particular quantity during month $i$. Moreover, we have also subscripted both the money

**Table 2. Effect of MFS transactions on monetary aggregates during a given month.**

| No | Transaction type | Transaction amount | Currency outside bank | M1 | M2 | TCSA |
|----|------------------|--------------------|-----------------------|-----|-----|------|
| (1) | (2) | (3) | (4) | (5) | (6) | (7) |
| 1 | Cash-in | $A$ | $-A$ | $+A \times MM_n$ | $+A \times MM_b$ | $+A$ |
| 2 | Cash-out | $B$ | $+B$ | $-B \times MM_n$ | $-B \times MM_b$ | $-B$ |
| 3 | P2P | $C$ | – | – | – | – |
| 4 | B2P | $D$ | – | – | – | $+D$ |
| 5 | P2B | $E$ | – | – | – | $-E$ |
| 6 | Merchant payment | $F$ | – | – | – | – |
| 7 | Government payment | $G$ | – | – | – | $+G$ |
| 8 | Inward remittances | $H$ | – | – | – | $+H$ |
| Total: | - - | $A + B + C + D + E + F + G + H$ | $B - A$ | $(A - B) \times MM_n$ | $(A - B) \times MM_b$ | $A - B + D - E + G + H$ |

**Table 3. Cumulative effect of MFS transactions on monetary aggregates over the year.**

| No | Transaction type | Transaction amount | Currency outside bank | M1 | M2 | TCSA |
|----|------------------|--------------------|-----------------------|-----|-----|------|
| (1) | (2) | (3) | (4) | (5) | (6) | (7) |
| 1 | Cash-in | $\sum_{i=1}^n A_i$ | $-\sum_{i=1}^n A_i$ | $\sum_{i=1}^n A_i \times MM_{n_i}$ | $\sum_{i=1}^n A_i \times MM_{b_i}$ | $\sum_{i=1}^n A_i$ |
| 2 | Cash-out | $\sum_{i=1}^n B_i$ | $\sum_{i=1}^n B_i$ | $-\sum_{i=1}^n B_i \times MM_{n_i}$ | $-\sum_{i=1}^n B_i \times MM_{b_i}$ | $-\sum_{i=1}^n B_i$ |
| 3 | P2P | $\sum_{i=1}^n C_i$ | – | – | – | – |
| 4 | B2P | $\sum_{i=1}^n D_i$ | – | – | – | $\sum_{i=1}^n D_i$ |
| 5 | P2B | $\sum_{i=1}^n E_i$ | – | – | – | $-\sum_{i=1}^n E_i$ |
| 6 | Merchant payment | $\sum_{i=1}^n F_i$ | – | – | – | – |
| 7 | Government payment | $\sum_{i=1}^n G_i$ | – | – | – | $\sum_{i=1}^n G_i$ |
| 8 | Inward remittance | $\sum_{i=1}^n H_i$ | – | – | – | $\sum_{i=1}^n H_i$ |
| Total: | | $\sum_{i=1}^n (A_i + B_i + C_i + D_i + E_i + F_i + G_i + H_i)$ | $\sum_{i=1}^n (B_i - A_i)$ | $\sum_{i=1}^n (A_i - B_i) \times MM_{n_i}$ | $\sum_{i=1}^n (A_i - B_i) \times MM_{b_i}$ | $\sum_{i=1}^n (A_i - B_i + D_i - E_i + G_i + H_i)$ |

multipliers as they do not remain constant over the months. The cumulative impacts of MFS operations on different monetary aggregates are algebraically represented in Table 3.

## 5 Methodology

Our empirical estimation is carried out in multiple steps:

- In the first step of our analysis, we calculate both the narrow and broad money multipliers, namely, $MM_n$ and $MM_b$ on monthly basis. To do so, we collect monthly data of total amount of narrow money (M1), broad money (M2) and monetary base (MB). Then we divide M1 by MB to calculate the narrow money multiplier ($MM_n$) for the respective month. On the other hand, to calculate broad money multiplier ($MM_b$), we divide monthly quantity of total broad money (M2) by the respective monetary base (MB).

- Next, we collect consolidated monthly data of cash-in (A), cash-out (B), P2P transaction (C), B2P transaction (D), P2B transaction (E), merchant payment (F), government payment (G) and inward remittance (H) carried out through mobile financial services.

- After that, we use the formula presented at the last row of column 4 in Table 2, i.e., (B-A) to capture the monthly change in currency outside bank resulting from the MFS transactions. Currency outside bank is an important monetary indicator as it represents physical money in people's wallets and these are the monies that are most directly spent on purchasing goods and services. If too much paper money enters into people's pockets due to MFS transactions, then they happen to spend more cash in purchasing goods and services which may possibly result into a rise in general price level. On the other hand, if currency outside bank is reduced due to MFS transactions, then people have less cash in their pockets. So, they tend to spend less cash on goods and services which may slow down the process of price hike. To what extent the price level will respond to the changes in currency outside bank resulting from MFS transactions is beyond the scope of this study. Here, we are contained with the quantification of changes in currency outside bank that has been brought about by the MFS transactions throughout the month.

- Till now, we have gathered monthly data of all types of MFS transactions and we have also calculated both the narrow and broad money multipliers. Moreover, in the last step, we have estimated the change in currency outside bank as an eventual consequence of MFS transactions. So, now we can use the formulas presented at column 5 and 6 of Table 2 to measure

the monthly changes in narrow money and broad money respectively arising from transactions through the mobile applications.

- After we have appraised the monthly changes in currency outside bank, narrow money and broad money, we are now in the position to assess its annual magnitude. We now use formulas presented at the last row of column 4, 5 and 6 of Table 3 to gauge the annual changes in currency outside bank, narrow money (M1) and broad money (M2) that have been brought about by the mobile financial transactions occurring throughout the country during the entire year.

## 6 Data

Here, in the first place, we gather information regarding the total volume of MFS transactions occurred in Bangladesh during the time span of January 2018 to January 2021. The span is selected depending upon the availability of the data. To be precise, Bangladesh Bank, the central bank of Bangladesh, starts to publish monetary data (both the narrow and broad money) on monthly basis starting from January 2018 and at the time of writing this piece the latest available update is for January 2021 [30]. Other public database like World Bank Open Data reports broad money of Bangladesh for a longer period of time. However, World Bank has discontinued the narrow money series (previously FM.LBL.MONY.CN) for long [31]. Moreover, OECD database does not report Bangladeshi monetary data [20]. So, we only have the time series data of Bangladeshi narrow money in the range January 2018 to January 2021 from the reliable sources. On the contrary, we use total currency issuued by the central bank as a measure of base money as it matches the exposition presented here and it is available as part of the issue department balance sheet of Bangladesh Bank for a longer period of time [17]. Meanwhile, as the history of MFS in Bangladesh is quite new, we have a shorter range of MFS data available at hand. To date, MFS data are obtainable only from December 2016 to May 2021 on a monthly basis from Bangladesh Bank [24]. For our analysis, we need data of narrow money, broad money, monetary base and MFS transactions all in a coherent manner, i.e., we need to truncate the longer series to the extent of the shorter ones. The common time span amongst which all these data series are available is from January 2018 to January 2021, i.e., we have 37 (thirty seven) months of data at hand to carry out our empirical analysis. The selected time span also coincidentally reflects a period of remarkable growth of mobile financial services in Bangladesh.

In the first step, we calculate $MM_n$ and $MM_b$ by dividing the monthly value of M1 and M2 by the corresponding monetary base. Estimation results are presented in Fig 1. From Fig 1, it is evident that the narrow money multiplier ($MM_n$) effectively remains constant throughout the time of our analysis. Precisely, its average value is 1.59 while the maximum and minimum are 1.66 and 1.50 respectively. So, very minuscule jittering is noticed in $MM_n$ values over the course of time and it moves very little above and below its mean. Like the narrow money multiplier ($MM_n$), broad money multiplier ($MM_b$) also varies very little during the period of investigation. Its average value is found to be 7.19 while its maximum and minimum values are 7.54 and 6.53 respectively.

In the next step, we estimate the amount of monthly changes in currency outside bank as a result of mobile financial transactions using the formula shown in column 4 of the last row of Table 2. We also calculate the annual changes in currency outside bank by the formula stated in column 4 of the last row of Table 3. The estimated values of changes in currency outside bank both in monthly and annual basis are illustrated in Fig 2. From the left segment of Fig 2, we can observe that the monthly changes of currency outside bank are mostly negative which

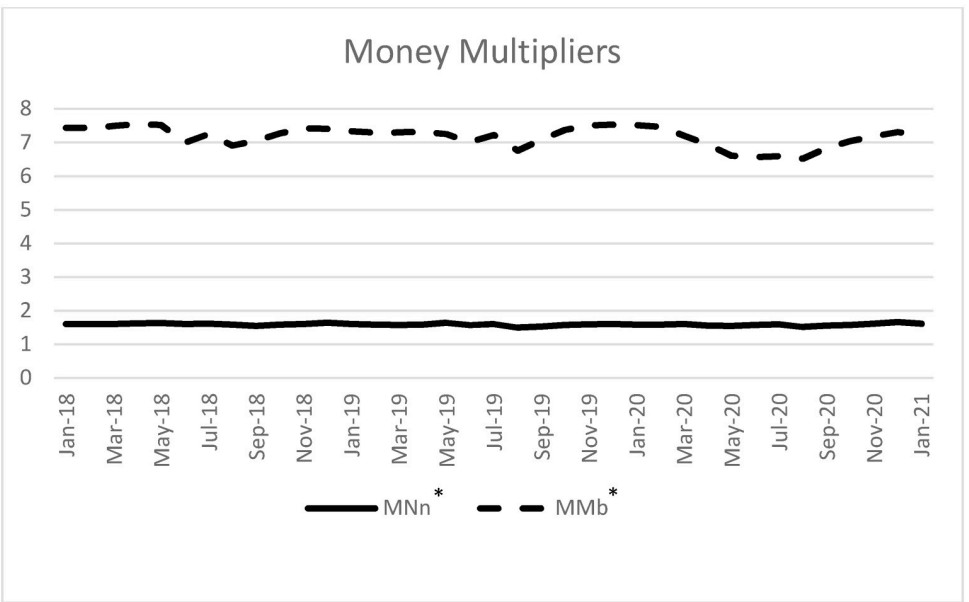

**Fig 1. Narrow money multiplier ($MM_n$) and broad money multiplier($MM_b$).** * We use total currency issued by the central bank as a measure of base money to match the exposition presented here.

implies that more and more physical currencies are entering into the banking system through the trust cum settlement account (TCSA) accounts of MFS providers although we have also seen some short-lived positive spikes in the series. These short spikes indicate outflow of currency from banking system (through the TCSA accounts) to public. However, in spite of these short positive spikes, annual contributions of MFS transactions on currency outside bank are overly negative, i.e., cash currencies are entering into the banking channel and the

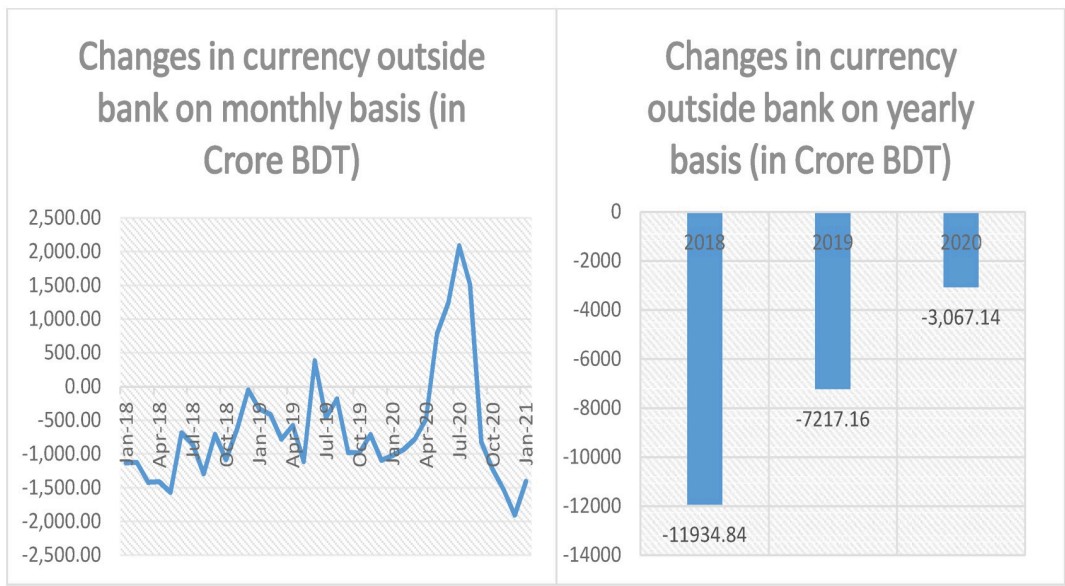

**Fig 2. Monthly and yearly changes in currency outside bank due to MFS transactions.**

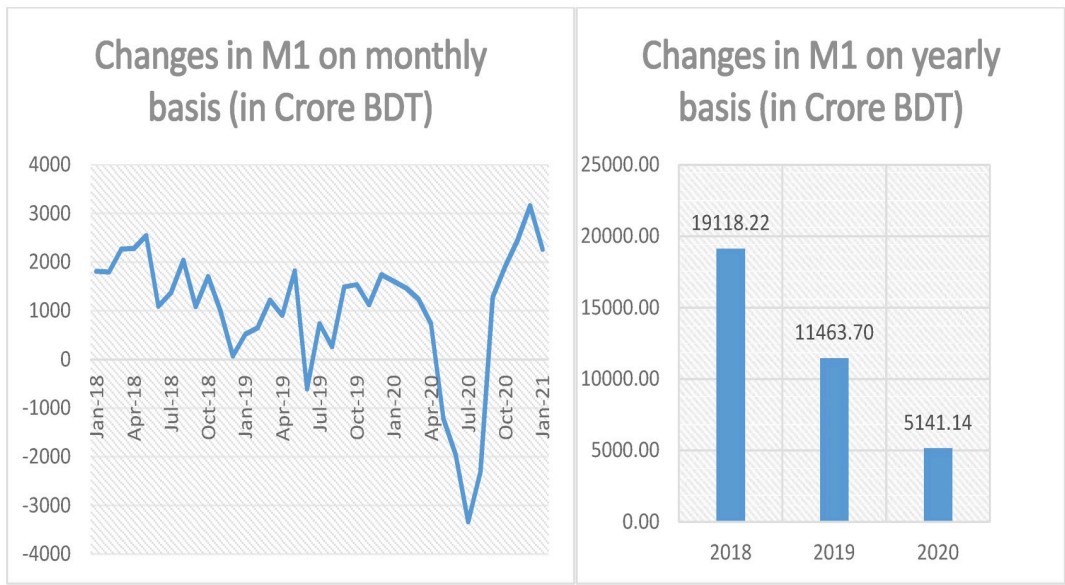

**Fig 3. Monthly and yearly changes in narrow money (M1) due to MFS transactions.**

corresponding data are shown on the right hand side of Fig 2. From the right segment of Fig 2, we can see that during 2018, 2019 and 2020, the changes in currency outside bank due to MFS operations are roughly BDT -11,934.84, -7,217.16 and -3,067.14 crore respectively which implies during these periods, BDT 11,934.84, 7,217.16 and 3,067.14 crore of physical currencies (after appropriate netting) have entered into the banking system through the MFS channel. We have seen a declining annual trend in the value of changes in currency outside bank originating from MFS transactions. We will briefly explain this fact at the end of this section.

As new money has entered into the banking system through MFS, they (this newly entered money) will then be amplified inside the bank according to the theory of fractional reserve banking. We will gauge the extent of narrow money and broad money created by this newly entered money. To measure the monthly and annual changes in narrow money brought about by the MFS transactions, we will use the formulas presented at column 5 of the last row of Tables 2 and 3 respectively. The results are exhibited in Fig 3. From the left hand side of Fig 3, we can see that the monthly changes in narrow money are roughly positive throughout the period under investigation with a few negative prongs. In spite of having a few negative prongs, the annual changes in narrow money, estimated by the formula presented on column 5 at the last row of Table 3, are mostly positive and these results are depicted on the right hand side of Fig 3. The right segment of Fig 3 demonstrates that during 2018, 2019 and 2020, approximately BDT 19,118.22, 11,463.70 and 5,141.14 crore of narrow money has been created by the mobile transactions. These newly created narrow monies comprise 5.82%, 3.49% and 1.57% of the total narrow money in circulation as on January 2021. Again, a declining trend is noticed in the amount of newly created narrow money. We will briefly explain this at the end of the data section.

Meanwhile, the monthly and annual changes in broad money brought about by the MFS transactions are calculated using the formulas presented at column 6 of the last row of Tables 2 and 3 respectively and the results are demonstrated in Fig 4. As anticipated, the monthly creation of broad money through MFS transactions is roughly positive with a few exceptions as can be seen from the left hand side of Fig 4. But, in spite of having some negative tips along the

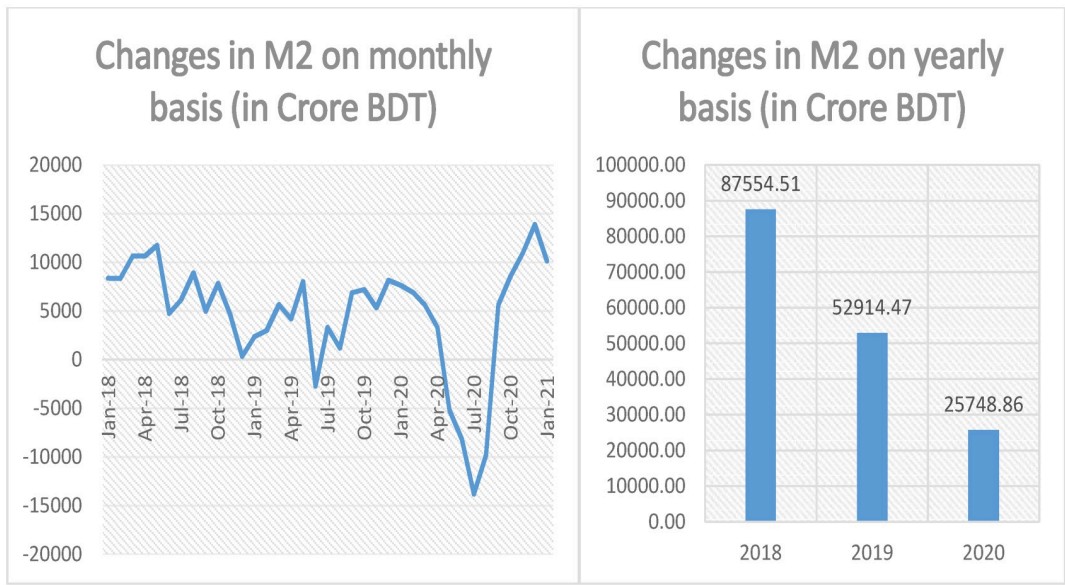

**Fig 4. Monthly and yearly changes in broad money (M2) due to MFS transactions.**

way, the cumulative amount of broad money created on an annual basis shows a steady, positive and declining trend (see right hand side of Fig 4). In the year 2018, 2019 and 2020, approximately BDT 87,554.51, 52,914.47 and 25,748.86 crore amount of broad money has been created from MFS operations which constitutes 5.94%, 3.59% and 1.75% of the total broad money supply as on January 2021. Again, we show a declining trend in the creation of broad money by various modes of MFS transactions over the years.

From the above discussion, it is evident that the changes in all three monetary variables, namely, currency outside bank, narrow money and broad money stemming from MFS transactions have been showing declining trends for the last couple of years. In fact, all the three variables exhibit the largest amount of reverse spikes during the period May 2020 to August 2020. The above fact is clearly depicted in Illustration: Figs 2–4 and it is not hard to remember that COVID-19 pandemic first landed its feet in Bangladesh on 8th of March 2020 [32]. To combat the rapidly worsening situation, government of Bangladesh declared a state-wide lockdown starting from 26th of March 2020 [33]. The complete lockdown was then subsequently enhanced up to 30th of May 2020 [34]. Although the lockdown was partially lifted after 30th of May 2020, public movements were then largely restricted up until 1st of September 2020 [35]. Country-wide complete lockdown and movement restrictions eventually take its toll on the economy and according to a survey conducted by Power and Participation Research Centre and BRAC Institute of Governance and Development brings the bitter truth in the forefront: Per capita daily income of urban and rural poor has been reduced by nearly 80% by the ravaging global pandemic followed by complete shut-down of economic activity and subsequent movement restrictions [36]. As people's income is substantially reduced, they tend to dig up their savings, i.e., they cash-out most of their savings for personal consumption and this explains the reverse spikes observed in the data. The reverse spikes against the trends in currency outside bank and monetary aggregates are predominantly due to this excessive, unnatural amount of cash-outs made by MFS customers facing an unprecedented global epidemic.

## 7 Discussion and policy implication

In recent times, the government of Bangladesh has taken decisions to curb interest rate in order to *reinvigorate* business activities around the country. Finance minister of Bangladesh after attending a meeting with the directors and Chief Executive Officers (CEOs) of the scheduled banks on December 2019 declared that the maximum lending rate, apart from the credit card lending, should be no more than 9% while for deposits, it could be as high as 6%. Finance minister also vowed that the new rates would come into effect from 1st of April 2020 [37]. Bangladesh Bank, the central bank of Bangladesh, subsequently issued a circular to all the scheduled banks operating in Bangladesh regarding the new interest rate cap [38]. The above declaration of rationalization of lending rate came coincidentally at the height of COVID-19 pandemic which was supposed to ease the financials of the businesses. However, instead of setting regulatory caps on the interest rate, the government could equivalently use a number of monetary policy tools to achieve the same objective. These tools include among others bank rates, open market operations, repo and reverse repo, Cash Reserve Requirement (CRR) and Statutory Liquidity Requirement (SLR) [39]. Nevertheless, the government rather took a direct route to cap interest rate through an edict. Very recently, such regulatory capping of interest rate was carried out in Kenya [40] and the results of this initiative were not very promising. In September 2016, the lending rate in Kenya was reset to at most 4% above the central bank rate while the deposit rate was readjusted to minimum 70% of the central bank rate [40]. Numerous studies have shown that the capping of interest rate had negatively influenced banks' performances in Kenya [40, 41] while a meagre 0.2% growth in credit was attained in return [42]. Moreover, some studies even suggested that interest rate capping had substantially reduced the profit margins and lending volumes of the commercial banks which eventually resulted into a huge number of employee lay-off [43]. Thus, the economic costs of interest rate capping were heavy in Kenya while the benefits were not so impressive as anticipated. Here, we argue that the initiative to rejuvenate business activity through artificially cutting interest rate, is inherently self-contradicting. Reducing the lending rate, can only increase loans and advances when the banks/financial institutions have spare capacity, i.e., idle funds (in absence of spare capacity, banks cannot disburse new loans whatever the interest rate and demand for credit may be as they have no loanable funds). So, only when the banks have idle funds, it can be effectively lent out. However, if the banks have idle money in the first place, then the deposit rate decreases automatically following the law of demand and supply and the lending rate follows its trail. Thus, any manual intervention of interest rate capping to boost up credits is not very much feasible and the empirical evidence from Kenya reinforces the reasoning. To us, financial inclusion, i.e., bringing the unbanked community into banks/MFS, can, to some extent, serve the purpose of reducing market interest by increasing the money supply and for this, MFS is the best option at hand as it has its footmark even in the furthest corner of the country. As we have discussed in the previous sections, MFS transactions have created a substantial amount of money through the process of fractional reserve banking. An obvious consequence of the enhancement of money supply is the reduction of interest rate as interest rate is simply the cost of borrowing money. When there is available money, its cost (interest rate) decreases whereas when the money is scarce, its cost soars. Thus, according to the discussion presented above, we believe that the government, instead of capping the interest rate, should strive even harder (than before) to bring the unbanked community under MFS as it is the most widely spread financial institution found even in the remotest corner of the country. If more and more people are included into the mobile financial system, money supply will obviously increase and interest rate will fall as an inevitable consequence.

## 8 Future work

Current study can be reasonably extended to a number of different directions. For example, although we have substantiated the money creation process through MFS transactions in the context of Bangladesh, we did not analyze the relative inclusivity of such a system, i.e., whether both the urban and rural population participate equally in the money creation process or it is the hegemony of the urban population only. Besides, the role played by different economic agents, e.g., regulators, MFS providers, merchants, agent points, end users etcetera in the process and the network dynamics of the whole ecosystem are yet to be investigated. Moreover, although we have quantified the impact of MFS transactions on the overall money supply, the responsiveness of different monetary aggregates to changes in mobile money is left unattended for the purpose of current study. A more econometric paper may resort to analyze the responsiveness of narrow money (M1), broad money (M2) and other aggregates alike to changes in e-money possibly under a structural VAR framework which is beyond the scope of the current study.

## 9 Conclusion

All the financial services provided by the MFS except for cash-in and cash-out can be carried out from home by simply using a smart phone which means MFS providers are rendering 24/7, one-stop, home services with zero waiting time to their respective clienteles. Moreover, due to country-wide coverage of the mobile network, MFS providers have access to the remotest corner of the country where banks can barely open and operate a branch as long as profitability is concerned. Due to ease of use, diversity of services and availability of mobile networks, usages and applications of MFS are increasing at a pace greater than ever before and through the process, a huge amount of hitherto *unbanked* deposits are added to the formal monetary aggregates of the country. Here, we try to quantify the impact of this newly added money on the overall money supply of Bangladesh. In the first place, we derive specific formulas relating MFS transactions with currency outside banks, M1 and M2. Then using these formulas, we empirically estimate the contribution of mobile money on different monetary aggregates on monthly basis and also in yearly cumulative forms. To our surprise, we have found that BDT 22,219.14 crore of previously informal money/deposit has entered into the banking system through MFS transactions during 2018-2021 which is nearly 10.93% of total currency issued up until January 2021. The money thus entered into the banking system will then be amplified according to the theory of money multiplier and it has been observed that BDT 35,723.06 crore of narrow money and BDT 1,66,217.85 crore of broad money have been created on the process during the aforementioned time. This newly created narrow money and broad money comprise 10.88% and 11.29% respectively of the total M1 and M2 at the point of January 2021. As the MFS operations are contributing heavily to the monetary aggregates, they have the potential to maneuver interest rate also as monetary aggregates and interest rates are closely related. As the MFS can significantly influence two important monetary anchors, i.e., money supply and interest rate, special emphasis should be given to the proliferation of MFS across the country during the formulation of half-yearly monetary policy of Bangladesh. If the government needs to readjust/reduce the interest rate with an intent to stimulate economic growth, it, instead of explicitly capping interest rates, can alternatively enhance the coverage of mobile financial services throughout the country to effectively add new money to the existing monetary aggregates and in the process, can readjust interest rate to an intended lower level.

## Acknowledgments

**Disclaimer**: The views and opinions expressed in this article are those of the author and do not necessarily reflect the official policy or position of Bangladesh Bank in this regard.

## Author Contributions

**Conceptualization:** Ahmed Mehedi Nizam.

**Data curation:** Ahmed Mehedi Nizam.

**Formal analysis:** Ahmed Mehedi Nizam.

**Investigation:** Ahmed Mehedi Nizam.

**Methodology:** Ahmed Mehedi Nizam.

**Resources:** Ahmed Mehedi Nizam.

**Software:** Ahmed Mehedi Nizam.

**Supervision:** Ahmed Mehedi Nizam.

**Validation:** Ahmed Mehedi Nizam.

**Visualization:** Ahmed Mehedi Nizam.

**Writing – original draft:** Ahmed Mehedi Nizam.

**Writing – review & editing:** Ahmed Mehedi Nizam.

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
