## [Decision Letter · Decision Letter 0]

22 Feb 2022

PONE-D-21-28479Impact of e-money on money supply: Estimation and policy implication for BangladeshPLOS ONE

Dear Dr. Nizam,

Thank you for submitting your manuscript to PLOS ONE. After careful consideration, we feel that it has merit but does not fully meet PLOS ONE’s publication criteria as it currently stands. Therefore, we invite you to submit a revised version of the manuscript that addresses the points raised during the review process. In your resubmission, please make sure you address both comments from the reviewer.

We look forward to receiving your revised manuscript.

Kind regards,

Federico Botta

Academic Editor

PLOS ONE

Journal Requirements:

Whilst you may use any professional scientific editing service of your choice, PLOS has partnered with both American Journal Experts (AJE) and Editage to provide discounted services to PLOS authors. Both organizations have experience helping authors meet PLOS guidelines and can provide language editing, translation, manuscript formatting, and figure formatting to ensure your manuscript meets our submission guidelines. To take advantage of our partnership with AJE, visit the AJE website (http://aje.com/go/plos) for a 15% discount off AJE services. To take advantage of our partnership with Editage, visit the Editage website (www.editage.com) and enter referral code PLOSEDIT for a 15% discount off Editage services.  If the PLOS editorial team finds any language issues in text that either AJE or Editage has edited, the service provider will re-edit the text for free.

Reviewers' comments:

Reviewer's Responses to Questions

**Comments to the Author**

1. Is the manuscript technically sound, and do the data support the conclusions?

Reviewer #1: Yes

2. Has the statistical analysis been performed appropriately and rigorously? 

Reviewer #1: Yes

3. Have the authors made all data underlying the findings in their manuscript fully available?

Reviewer #1: Yes

4. Is the manuscript presented in an intelligible fashion and written in standard English?

Reviewer #1: Yes

5. Review Comments to the Author

Reviewer #1: I would like to thank the authors for the effort and quality of the manuscript.

The paper proposes an analysis of Bangladesh money supply, and in particular of the impact that the introduction of Mobile Financial Services (MFS) have had on it in recent years. MFS represent a recent and rapidly increasing innovation, which is now massively used in Bangladesh, allowing people and merchants to make payments and transactions with e-money through the use of mobile phones. E-money can only be issued by licensed entities, in exchange for an equal amount of legal tenders.

The authors show the big impact MFS have had on Bangladesh money supply since their introduction, contributing to around 10% of relevant quantities between 2018 and 2021, and argue that such increase in money supply show how an increase in MFS coverage across the country (especially in rural parts), could effectively work better at stimulating economic growth than artificially capping interest rates.

The manuscript has a clear and well-paced exposition. In particular, I appreciated the space reserved to the historical and national context of Bangladesh, both in terms of financial and monetary interventions, but also banking and internet coverage, in the manuscript and in particular introduction and paragraph 3. Moreover, the author also spends dedicate space to carefully introducing all relevant economic terms and definitions in two dedicated paragraphs (2 and 4), which greatly improve the readability of the manuscript, avoiding possible confusion and making each claim in the manuscript clear and narrow. In particular, this also allows a possibly more interdisciplinary audience to understand a more economics paper as this one, which I find particularly suited to this journal.

The analysis per-se can be considered quite simple, as the data and estimates only involve the computation and analysis of different aggreagate numbers. However, it is my opinion that the overall paper, in terms of national context, extended explanation of the necessary terms and definitions, detailed analysis of the different trends in terms of the national context, and discussion of these in light of different economic policies, provide a complete enough work to be considered for publication in Plos One.

I therefore recommend the acceptance of this paper for publication in PLOS ONE.

I just want to highlight a couple minor comments, which would slightly improve the paper in my opinion, but do not prevent me from recommending acceptance:

1. the abstract should reserve more space to the quantitative results (e.g. the ~10% of money supply due to MFS), and a bit less space to the qualitative argument against capping interest rates, to reflect the importance of the results and the space reserved in the manuscript to both

2. Some space could be dedicated in the discussion to limitations or possible extension of the work. In particular, a natural extension of this study, to better understand the dynamics and importance of MFS transactions, could be to study the network of these MFS transactions, to better understand the role played by different kinds of agents (merchants, people ecc) and the effective inclusivity of such system (how many rural people are actually active, what's the distribution of total money in accounts etc.).

6. PLOS authors have the option to publish the peer review history of their article (what does this mean?). If published, this will include your full peer review and any attached files.

Reviewer #1: No

---

## [Author Response · Author response to Decision Letter 0]

17 Mar 2022

Reviewer’s comment #1: 

The abstract should reserve more space to the quantitative results (e.g. the ~10% of money supply due to MFS), and a bit less space to the qualitative argument against capping interest rates, to reflect the importance of the results and the space reserved in the manuscript to both.

Answer: 

We have rewritten the abstract according to the reviewer’s suggestion. It now reads as follows: 

With the rapid proliferation of mobile telephony and the establishment of an IT-enabled payment and settlement system, Bangladesh nowadays is experiencing a remarkable growth in the usage of mobile financial services (MFS). As more and more people are opting to use this service, a huge number of mobile accounts are opened every day and a substantial amount of money is deposited, withdrawn and transferred frequently through the mobile network. This ever-increasing amount of mobile money flowing through the network may have a sizeable impact on the overall money supply of the country. Thus far, no systematic study has been conducted to quantify the impact of the mobile money on the conventional money supply of Bangladesh. In this study, we attempt to quantify the contribution of mobile money on the money supply which is an important quantity-based nominal anchor of monetary policy in Bangladesh. Apart from deriving algebraic relationships between money supply and e-money, here we have empirically shown that during the 03 years span of 2018-2021, MFS transactions account for nearly 10.88% and 11.29% of total narrow and broad money supply of Bangladesh as on January 2021. Besides, we also qualitatively discuss the impact of e-money on an important price-based nominal anchor of monetary policy in Bangladesh, i.e., interest rate. Based upon the above discussion, here we argue that MFS can act as an effective tool to slash interest rate by a reasonable proportion through adding significantly to the overall supply of money in Bangladesh.

Reviewer’s comment #2: 

Some space could be dedicated in the discussion to limitations or possible extension of the work. In particular, a natural extension of this study, to better understand the dynamics and importance of MFS transactions, could be to study the network of these MFS transactions, to better understand the role played by different kinds of agents (merchants, people ecc) and the effective inclusivity of such system (how many rural people are actually active, what's the distribution of total money in accounts etc.). 

Answer: 

We have added a separate section titled ‘Future Work’ which reads as follows: 

Current study can be reasonably extended to a number of different directions. For example, although we have substantiated the money creation process through MFS transactions in the context of Bangladesh, we did not analyze the relative inclusivity of such a system, i.e., whether both the urban and rural population participate equally in the money creation process or it is the hegemony of the urban population only. Besides, the role played by different economic agents, e.g., regulators, MFS providers, merchants, agent points, end users etcetera in the process and the network dynamics of the whole ecosystem are yet to be investigated. Moreover, although we have quantified the impact of MFS transactions on the overall money supply, the responsiveness of different monetary aggregates to changes in mobile money is left unattended for the purpose of current study. A more econometric paper may resort to analyze the responsiveness of narrow money (M1), broad money (M2) and other aggregates alike to changes in e-money possibly under a structural VAR framework which is beyond the scope of the current study.

---

## [Decision Letter · Decision Letter 1]

12 Apr 2022

Impact of e-money on money supply: Estimation and policy implication for Bangladesh

PONE-D-21-28479R1

Dear Dr. Nizam,

We’re pleased to inform you that your manuscript has been judged scientifically suitable for publication and will be formally accepted for publication once it meets all outstanding technical requirements.

Kind regards,

Federico Botta

Academic Editor

PLOS ONE

Additional Editor Comments (optional):

Reviewers' comments:

Reviewer's Responses to Questions

**Comments to the Author**

1. If the authors have adequately addressed your comments raised in a previous round of review and you feel that this manuscript is now acceptable for publication, you may indicate that here to bypass the “Comments to the Author” section, enter your conflict of interest statement in the “Confidential to Editor” section, and submit your "Accept" recommendation.

Reviewer #1: All comments have been addressed

2. Is the manuscript technically sound, and do the data support the conclusions?

Reviewer #1: Yes

3. Has the statistical analysis been performed appropriately and rigorously? 

Reviewer #1: Yes

4. Have the authors made all data underlying the findings in their manuscript fully available?

Reviewer #1: Yes

5. Is the manuscript presented in an intelligible fashion and written in standard English?

Reviewer #1: Yes

6. Review Comments to the Author

Reviewer #1: I wanna thank the authors for addressing my previous minor comments.

I am happy to confirm my previous review and recommend the paper for publication, and look forward to further work using this interesting dataset.

7. PLOS authors have the option to publish the peer review history of their article (what does this mean?). If published, this will include your full peer review and any attached files.

Reviewer #1: No

---

## [Editor Report · Acceptance letter]

18 Apr 2022

PONE-D-21-28479R1 

Impact of e-money on money supply: Estimation and policy implication for Bangladesh 

Dear Dr. Nizam:

I'm pleased to inform you that your manuscript has been deemed suitable for publication in PLOS ONE. Congratulations! Your manuscript is now with our production department. 

Kind regards, 

on behalf of

Dr. Federico Botta 

Academic Editor

PLOS ONE